# Fabrication of Multilayered Biofunctional Material with an Enamel-like Structure

**DOI:** 10.3390/ijms232213810

**Published:** 2022-11-09

**Authors:** Yu Yuan Zhang, Quan Li Li, Hai Ming Wong

**Affiliations:** 1Faculty of Dentistry, The University of Hong Kong, 34 Hospital Road, Hong Kong 000000, China; 2Collage and Hospital of Stomatology, Anhui Medical University, No. 69, Meishan Road, Heifei 230000, China

**Keywords:** graphene oxide, hierarchical structure, enamel-like structure, enamel-inspired material, biofunctional material

## Abstract

The oral cavity is an environment with diverse bacteria; thus, antibacterial materials are crucial for treating and preventing dental diseases. There is a high demand for materials with an enamel-like architecture because of the high failure rate of dental restorations, due to the physical differences between dental materials and enamel. However, recreating the distinctive apatite composition and hierarchical architecture of enamel is challenging. The aim of this study was to synthesize a novel material with an enamel-like structure and antibacterial ability. We established a non-cell biomimetic method of evaporation-based bottom-up self-assembly combined with a layer-by-layer technique and introduced an antibacterial agent (graphene oxide) to fabricate a biofunctional material with an enamel-like architecture and antibacterial ability. Specifically, enamel-like graphene oxide-hydroxyapatite crystals, formed on a customized mineralization template, were assembled into an enamel-like prismatic structure with a highly organized orientation preferentially along the *c*-axis through evaporation-based bottom-up self-assembly. With the aid of layer-by-layer absorption, we then fabricated a bulk macroscopic multilayered biofunctional material with a hierarchical enamel-like architecture. This enamel-inspired biomaterial could effectively resolve the problem in dental restoration and brings new prospects for the synthesis of other enamel-inspired biomaterials.

## 1. Introduction

Native hard tissue, such as nacre, bone, and dental enamel, displays excellent stiffness and toughness, owing to its multiscale hierarchical architecture [1]. The hierarchical architecture improves the biostability and functional performance of biomaterials, as well as conferring biocompatibility, by providing a suitable environment for cell attachment and growth. These characteristics are crucial in medical and dental engineering. Therefore, mimicking the nano- and micro-structure of native tissue in macroscopic artificial materials is one of the most active areas within biomaterials research. Notable examples include materials that mimic nacre’s [2] uniform nano- and micro-structure [3,4]. 

Dental enamel is the hardest human tissue and is composed of repeating prismatic units. Each prism is 4–8 µm in diameter and is made up of nanoscale hydroxyapatite (HA) with a cross-section of 25–100 nm aligned along the *c*-axis [5]. This distinct hierarchical structure gives enamel its excellent mechanical properties. Because enamel has no mechanism of cellular regeneration, it is rarely self-repaired when damaged. Artificial materials are commonly used to restore dental hard tissue lesions in dental clinics. However, restoration failure, which includes fractures and marginal leakage caused by the physical mismatch between restorative materials and native enamel, remains a problem [6]. Studies have shown that artificial materials with enamel-like structures have better biocompatibility than those with randomly oriented structures [7,8,9].

Different approaches have been taken to synthesize enamel-like structures, including the addition of organic matrixes [10,11,12,13,14,15,16,17], the stirring synthesis of crystalline paste [18], and protein-induced biomineralization (for example, using amelogenin or elastin-like recombinant protein) [19,20,21]. Unfortunately, none of these approaches have satisfactorily recapitulated the microstructure of enamel. More precisely, no highly organized nanocrystal has been created that has the distinctive hierarchical architecture of native enamel from the crystallographic scale to the nano-, micro-, and macro-scales. 

HA is the primary component in the vertebral skeleton and native enamel (95% by weight). HA-based biomaterials have excellent biocompatibility and osteoconductivity [22,23,24,25]. Although HA has been applied in medicine and dentistry for bone and tooth repair, its poor wear resistance and fracture toughness limit its load-bearing engineering applications [26,27]. Graphene oxide (GO), which has high mechanical strength and hydrophilicity, as well as antibacterial properties, good biocompatibility, and biostability, is widely regarded as the most promising candidate for nanoscale reinforcement in biomaterials [28,29]. Accordingly, GO−HA composites have garnered considerable attention [28,30,31]. For example, GO has been shown to effectively strengthen HA in vivo without any toxic effects [30]. Nevertheless, only GO−HA crystals with random interior structures have been obtained. GO–HA crystals with homogeneous hierarchical structures have not been previously fabricated [32]. 

To overcome the shortcomings of these previous attempts, we established a method based on a combination of layer-by-layer (LBL) deposition and evaporation-based bottom-up self-assembly. LBL deposition is a simple and efficient technique commonly applied to synthesize solid films with a controlled multilayer structure, whereby assembly can be based on electrostatic interactions, hydrogen bonding, charge transfer, covalent bonding, biological recognition, or hydrophobic interactions [33,34]. The bottom-up approach is an innovative and effective approach for fabricating bulk materials. In this technique, the physical or chemical interactions between particles are used to manipulate small building blocks to construct mesoscopic or macroscopic functional architectures with well-defined morphologies, shapes, and patterns [35,36]. Evaporation has been shown to further improve the bottom-up self-assembly of enamel-like crystals [37]. By combining evaporation-based bottom-up and LBL assembly in the present study, we were able to replicate the hierarchical architecture of enamel. The aim of this study was, therefore, to synthesize a novel material with an enamel-like structure and antibacterial ability to resolve the limitations of present dental materials.

## 2. Results and Discussion

In this study, we established a non-cell-based biomimetic strategy to fabricate a novel bioactive material to resolve the limitations of the conventional dental materials. The material demonstrates enamel-like structural, mechanical, and esthetic properties, as well as antibacterial abilities and biocompatibility.

The oral cavity is an environment in which pathogenesis and development of diseases are known to happen frequently. The physical and structural mismatch between dental restorative materials and native enamel can cause marginal leakage, resulting in recurrent caries and restoration failure. Dental enamel is characterized by its excellent intrinsic–extrinsic fracture toughening mechanism, due to its organized hierarchical structure. Synthesis of the biomaterial with an enamel-like structure and anti-bacterial ability may effectively resolve the problems in dental restoration. In amelogenesis, enamel organic matrixes play the role of a template to guide the minerals into parallelly aligned HA prisms with highly organized orientation [32,38]. Inspired by this process of native enamel formation, the templated-directed method has been proposed and applied to synthesize materials with a highly ordered orientation [39,40,41]. Designing an epitaxial template is arduous in the templated-directed method, due to the stringent requirements for the template surface structure. Epitaxial templates should have rigid and highly ordered surface structures for imposing lattice matching [42]. In addition, removing the template to obtain individual nanomaterials is challenging.

Our solution to this challenge was to establish a non-cell-based biomimetic strategy by using a customized polyethylene membrane coated with polydopamine. Polyethylene membranes are soluble in isoamyl acetate solution. Polydopamine has strong adhesive properties to various substrates, and its catecholamine moieties can bind with Ca^2+^, further attracting PO_4_^3−^ and triggering crystal nucleation [43,44]. HA formed with the aid of polydopamine is aligned to the *c*-axis and parallel to the polydopamine layer [45]. Those enamel-like GO−HA crystals continued to aggregate and assemble into a crystal layer with highly organized crystal orientation. The macroscopic, multilayered GO−HA crystal was then constructed via sequential growth of the GO−HA crystal layers, followed by LBL deposition of a polyelectrolyte matrix. Lastly, we isolated the bulk GO−HA crystals and obtained the synthesized material by dissolving the mineralization template in an isoamyl acetate solution.

### 2.1. Synthesis of GO 

FTIR evaluation of the synthesized GO showed a range of oxygen functionalities, such as hydroxyl and carbonyl groups (Figure 1a). The band at 3397.86 cm^−1^ was assigned to the OH stretching vibration of surface hydroxyl groups, and the band at 1715.31 cm^–1^ was characteristic of the C=O stretch of the carboxylic acid group. The C–O bonds were assigned to the peak at 1227.55 cm^−1^, representing the oxide functional groups in GO and confirming the successful carbonization of citric acid into GO [46,47].

GO has a unique atomic and electronic structure that consists of variable sp^2^ and sp^3^ fractions [48]. After being carbonated, the color of the citric acid solution became dark as a result of the structural change [49,50]. The UV-visible adsorption spectrum of GO had an adsorption peak around 230 nm and a small shoulder at 300 nm, corresponding to the π–π* transitions of aromatic C-C bonds and the n–π* transitions of C=O (Figure 2) [28]. In addition, because GO was obtained through carbonizing citric acid, the carbon concentration of GO was higher than that of citric acid (Table 1).

### 2.2. Formation of Enamel-like GO–HA Crystals 

We performed XRD phase analysis according to the Joint Committee on Powder Diffraction Standards (JCPDS) card number 09-0432. The structures and composition of the minerals formed in the absence and presence of GO were similar to those of HA, and they had a high degree of crystallinity. The XRD patterns of pure HA and the GO−HA crystal layer fabricated with different concentrations of GO are shown in Figure 2B. The diffraction peaks at 2*θ* of 25.9°, 31.8°, 32.9°, 34.2°, 39.8°, 49.5°, and 53.2° were indexed to the (002), (112), (300), (202), (310), (213), and (004) crystal planes, respectively. We also observed diffraction peaks (002) at 2*θ* = 25.9°, (211) at 2*θ* = 31.8°, (300) at 2*θ* = 32.9°, and (202) at 2*θ* = 34.2° from HA (Figure 2B(a)) in the GO−HA crystal layers with the addition of different concentrations of GO (Figure 2B(b,c)). The peak intensity of the (002) plane in the GO−HA crystal layer was higher with the addition of 0.2 wt% GO (Figure 2B(b)) than with the addition of 0.4 wt% GO (Figure 2B(c)), demonstrating that the degree of crystallinity of HA was higher at the higher GO concentration. Our observation that the introduction of GO promoted a high degree of crystallinity of HA agreed with those of previous studies [29,51]. Due to the loss of crystallographic order and the irregular arrays of atoms in three dimensions, no trace of GO was detected in the GO−HA crystal layer [52,53].

FTIR evaluation verified the presence of GO in the GO−HA crystal layers fabricated with different concentrations of GO. The FTIR spectrum of the HA crystal layer showed the characteristic bands of PO_4_^3−^ (at 1035.96 and 565.87 cm^−1^) (Figure 1B) [54]. The spectrum of GO–HA crystal layers (Figure 1C,D) consisted of vibrations from HA and GO. The strong absorption peaks at 1412.77 cm^−1^ corresponded to the carbonate ions present in the PO_4_ site in HA [55]. The bending vibrations of PO_4_ were confirmed by the intense peaks of the GO–HA crystal layers (at 603.82 and 568.16 cm^−1^ for 0.2 wt% GO; and 604.23 and 567.67 cm^−1^ for 0.4 wt% GO), and C–O vibrations were also observed (at 1038.77 and 1042.23 cm^−1^ when fabricated with 0.2 and 0.4 wt% GO, respectively). Moreover, hydroxyl functional groups were observed (at 3416.22 and 3423.33 cm^−1^ for 0.2 and 0.4 wt% GO, respectively), demonstrating the formation of GO–HA crystals.

EDS was used for elemental analysis. Table 2 shows the presence of carbon, oxygen, fluorine, phosphorous, and calcium elements in the HA and GO−HA crystal layers. Moreover, the distribution of these elements was uniform, indicating the homogeneous structure of the crystal layer. After conjunction with GO, the elemental compositions changed. The carbon content increased with the introduction of GO (7.86% and 9.25% for 0.2 and 0.4 wt% GO, respectively, compared to 1.84 wt% in the HA layer without GO). Moreover, GO integration slightly decreased the Ca/P ratio, which may be due to the increased carbon concentration. The Ca/P ratios of the HA crystal layer, the GO−HA crystal layer fabricated with the addition of 0.2 wt% GO and the GO−HA crystal layer fabricated with the addition of 0.4 wt% GO were 1.69, 1.63, and 1.57, respectively.

From the SEM images, we observed the positive effect of GO on crystal growth, as it mediated both the crystal morphology and size. Without GO, the HA had a hexagonal morphology with a diameter of approximately 2 μm (Figure 3A). After the introduction of GO, the crystal morphology changed from hexagonal into a spear-like structure, and the crystal size decreased from the micro- to nanoscale. Both the morphology and size of the crystals were similar to those of native enamel (Figure 3B).

### 2.3. Fabrication of an Enamel-like, Multilayered Microstructure

During the oven heating process, free Ca^2+^ ions in the metastable mineralization solution were attracted and bound to catecholamine moieties of polydopamine in the mineralization template, triggering crystal nucleation and further promoting crystal growth. With the formation and aggregation of more crystals, the surface of the template was fully mineralized and covered by a thin layer of GO−HA crystals (Figure 3C). Due to the existence of crystal voids, the GO−HA crystal layer had a loose interior structure (Figure 3D). We observed that part of the crystal blocks grown on the mineralization template started to split away after six days of oven heating; in this case, it was not possible to form a macroscopic crystal biomaterial. 

Although the organic content in native enamel is low (less than 5% by weight), it plays an important role in regulating crystal growth and orientation, as well as forming enamel with a sophisticated intrinsic−extrinsic fracture-resistant mechanism [5]. During fabrication, we introduced the organic content using the LBL technique. 

Chitosan and alginate have excellent biocompatibility, biodegradability, and antimicrobial activity, as well as low immunogenicity [26,56]. The amino residues of chitosan electrostatically bind with the carboxylic residues of alginate to form a polyelectrolyte matrix film. These films have been used widely in tissue engineering, drug delivery, and wound healing [27]. For example, Mao et al. synthesized artificial nacre via the introduction of chitosan and alginate complexes [57]. Inspired by their work, we also introduced chitosan and alginate as the organic constitutes in the present study.

Figure 4 shows the FTIR spectrum of chitosan, alginate, and the chitosan−alginate polyelectrolyte complex. In the spectrum of chitosan (Figure 4a), the broadband at 3292.18 cm^−1^ corresponded to the amine and hydroxyl groups; it also contained characteristic bands of amide II and amino groups at 1583.18 and 1031.82 cm^−1^, respectively [53,58]. For alginate (Figure 4b), the C−O−C stretching band at 1030.04 cm^−1^ was attributed to its saccharide structure. In the spectrum of chitosan−alginate (Figure 4c), the amide II peak intensified. The amide II peak in the absorption band of chitosan at 1583.18 shifted to 1568.00 cm^−1^ after reaction with alginate, and the stretching vibrations of −OH and −NH_2_ at 3292.18 shifted to 3359.22 cm^−1^ and broadened. These changes suggested that chitosan–alginate polyelectrolyte complexes were formed by carboxylic groups in alginate that electrostatically interacted with ammonium groups in chitosan [59].

The polyelectrolyte matrixes were mineralized after reaction with the ions in the metastable mineralization solution. Mineralized polyelectrolyte matrixes provide crystal nuclei, leading to the formation of new crystals and the overgrowth of the original crystals (as indicated by the arrow in Figure 5C). SEM revealed that the surface of the initial GO−HA crystal layer was covered with the mineralized polyelectrolyte matrix film (Figure 5A,B). In a vacuum environment, some polyelectrolyte matrixes infiltrated into the voids among crystals and were mineralized, resulting in the disappearance of voids (as shown by the rectangle in Figure 5D), which caused the bonding of neighboring crystals, ultimately promoting the formation of a large-area GO−HA film with a multilayered structure. Figure 6 shows the formation of a homogeneous and dense GO−HA crystal layer in which newly formed crystals tightly bound together with no crystal voids (Figure 6C). The transversal micrographs showed that spear-like crystals extended from the surface of the initial crystal layer and were densely packed along the *c*-axis; in addition, the crystallographic *c*-axis had the same orientation, perpendicular to the surface of the initial crystal layer (Figure 6A–C). The XRD spectrum of the GO−HA crystal layers showed a high-intensity 002 peak (Figure 2B(b,c)), further suggesting that the formed crystals were oriented along the *c*-axis. After the LBL deposition treatment, the newly formed crystals tended to spontaneously self-assemble into bundles of prisms, resembling the distinct hierarchical architecture of native enamel (Figure 6C, inset). During the mineralization process, the capillary force, van der Waal’s force, and hydrophobic force between adjacent crystals caused them to fuse together and form a well-aligned crystal layer. The interface between the two crystal layers showed tight agglomeration and fusion (Figure 6A). The thickness of the macroscopic GO−HA crystal that consisted of two and four crystal layers was 5.8 and 17.9 µm, respectively (Figure 6A,E). After 10 cycles of mineralization and LBL deposition, the multilayered material with an enamel-like microstructure had a thickness of 81.3 µm (Figure 6F). In the present study, an enamel-inspired biofunctional material was successfully synthesized, which is a promising candidate for use in biomaterial research, especially for dental restoration [60].

### 2.4. Mechanical Evaluation of Synthesized Materials 

Neither the nanohardness of the final synthesized materials with 0.2 wt% GO matrix ((3.170 ± 0.285) GPa; *p* = 0.1103; *t*-test) nor with 0.4 wt% GO matrix ((3.334 ± 0.175) GPa; *p* = 0.36; *t*-test) showed any significant difference to the nanohardness of native enamel ((3.472 ± 0.246) GPa; Figure 7A). The modulus of the synthesized materials with 0.2 wt% GO matrix ((75.172 ± 5.014) GPa; *p* = 0.02; *t*-test) and 0.4 wt% GO matrix ((79.878 ± 7.990) GPa; *p* = 0.02; *t*-test) was significantly higher than that of native enamel ((68.044 ± 2.235) GPa; Figure 7B). However, the increase in GO concentration did not cause any significant changes in the nanohardness (*p* = 0.30; *t*-test; Figure 7A) or modulus (*p* = 0.40; *t*-test; Figure 7B) of the synthesized materials. Wei et al. also synthesized a multilayered composite with an enamel-like structure, which demonstrated hardness and an ultra-high Young’s modulus comparable to natural enamel [61]. However, this multilayered composite is composed of rutile dioxide nanorods that are different from the main constituent of native enamel.

### 2.5. Antibacterial and Biocompatibility of the Synthesized Materials

Dental caries is a disease caused by microbes. S. mutans plays an important role in dental caries, and it can produce large amounts of organic acids, leading to a decrease in pH value and resulting in the loss of inorganics in tooth demineralization [62]. Therefore, it is essential to synthesize restorative materials with antibacterial capacity. GO is a low-cost and highly effective carbon nanomaterial that shows excellent antibacterial activity and minimal cytotoxicity [28]. In the present study, GO was used and combined with HA to produce a synthesized material with antibacterial ability. 

After incubation for 24 h, the quantity of adherent bacteria on the synthesized material with 0.2 wt% GO matrix ((45.4 ± 21.65) × 10^4^ CFU/mL) was significantly less than that on the synthesized material without the addition of GO matrix ((141.20 ± 47.55) × 10^4^ CFU/mL; *p* = 0.0034, *t*-test; Figure 8A). The quantity of adherent bacteria on the synthesized material with 0.4 wt% GO matrix ((30.2 ± 9.09) × 10^4^ CFU/mL, *p* = 0.1858, *t*-test; Figure 8A) was not significantly different from that on the synthesized material with 0.2 wt% GO matrix. 

A CCK-8 assay was performed after cell culture for seven days to evaluate the cytotoxicity of the synthesized materials. Neither the synthesized bioactive material with 0.2 wt% GO matrix (70.83 ± 11.721; *p* = 0.61, *t*-test) nor the synthesized bioactive material with 0.4 wt% GO matrix (68.51 ± 7.166; *p* = 0.2362, *t*-test) showed a significant difference in cell viability with the HA sheet (73.67 ± 8.260; Figure 8B). Therefore, these data confirmed the biosafety of the synthesized material. 

We used an effective acellular method to fabricate a macroscopic multilayered GO–HA material with a hierarchical enamel-like architecture and antibacterial abilities. This enamel-inspired biofunctional material recreated the distinctive hierarchical architecture of enamel from the crystallographic scale to the nano-, micro- and macroscale. Its compositions are similar to that of native enamel. This study offers a promising candidate for dental restoration. The computer-aided design/computer-aided manufacturing (CAD/CAM) technique can also be applied to form the exact morphology and size of the restoration needed, before it is inserted into the prepared cavity of the tooth.

## 3. Materials and Methods

### 3.1. Synthesis of GO 

Two GO solutions (0.4 and 0.8 wt%) were prepared separately by first heating 2 g and 4 g, respectively, of citric acid powder (Sigma-Aldrich, St. Louis, MO, USA) at 200 °C until the mixture became a black liquid. This liquid was then added to 500 mL of deionized water. Finally, the pH was adjusted to 5.5 with 1 M NaOH solution [63].

### 3.2. Preparation of a Metastable Mineralization Solution Containing GO

The 0.4 and 0.8 wt% GO solutions were separately mixed with 500 mL of metastable calcium phosphate solution (5.8 mM Ca^2+^, 3.5 mM PO_4_^3−^ and 1.17 mM F^−^) to form the metastable mineralization solution, containing 0.2 and 0.4 wt% GO, respectively. The pH of the mineralization solution was adjusted to 5.5 with 0.1 M HCl and 0.1 M NaOH. The solution was stored at 4 °C before use.

### 3.3. Preparation of Chitosan and Alginate Solutions 

Chitosan (0.1 wt%) solution was prepared by dissolving 0.5 g of chitosan (Sigma-Aldrich, St. Louis, MO, USA) in 1% *w*/*v* acetic acid solution (500 mL), and the pH was adjusted to 5.5 with 1 M NaOH solution. Alginate solution (0.1 wt%) was prepared by dissolving 0.5 g of alginate (Sigma-Aldrich, St. Louis, MO, USA) in deionized water (500 mL), and the pH was adjusted to 5.5 with 0.1 M acetic acid. 

### 3.4. Preparation of Dopamine-Coated Polyethylene Membranes 

Polyethylene membranes with a diameter of 45 mm were polished with 1500-grit silicon carbide paper and then ultrasonically cleaned with deionized water. The polished polyethylene membranes were immersed in a freshly prepared 2 mg/mL dopamine solution (Sigma-Aldrich, St. Louis, MO, USA) (10 mM Tris buffer, pH 8.5) at 37 °C in the dark. After immersion for 24 h, the membranes were cleaned ultrasonically for 10 min with deionized water three times and dried under nitrogen. 

### 3.5. Fabrication of Bulk Multilayered GO−HA Crystals

The dopamine-coated polyethylene membranes were used as the mineralization template. The template was treated with alternating mineralization and LBL deposition cycles to fabricate the bulk multilayered GO−HA crystal. 

#### 3.5.1. Mineralization Process

A freshly prepared mineralization template was placed in the prepared metastable mineralization solution and heated in an oven at 70 °C for 7 days.

#### 3.5.2. LBL Deposition 

After mineralization, the template was rinsed with deionized water and dried in a vacuum desiccator. LBL deposition was then performed. Briefly, the dried sample was immersed, alternating between freshly prepared 0.1 wt% chitosan and 0.1 wt% alginate solutions in a vacuum chamber for 10 min. This process was followed by thorough rinsing with deionized water between each layer, until a total of 10 layers had been assembled.

After LBL deposition, the sample was placed back into the metastable mineralization solution and heated in an oven at 70 °C to continue the next cycle of treatment. 

### 3.6. Isolation of Synthesized Material 

After 10 cycles of mineralization and LBL deposition, as described above, the mineralization template was removed by immersion in isoamyl acetate solution to leave the isolated bulk GO−HA crystal and obtain the synthesized biofunctional material.

### 3.7. Characterization and Assessments

Scanning electron microscopy (SEM)

SEM analysis was conducted using a Carl Zeiss Supra 40 field emission scanning electron microscope (2–5 kV, depending on the sample state). The SEM samples were coated with Au film for 30 s at a constant current of 30 mA before observation.

Energy dispersive spectroscopy (EDS)

EDS data were acquired by EDS (Hitachi S4800, Hitachi Ltd., Tokyo, Japan; FEI, Sirion 200, Philips, Hillsboro, OR, USA).

Fourier transform infrared spectroscopy (FTIR)

Infrared spectra of the samples were acquired by a Thermo Scientific Nicolet 8700 FT-IR (Thermo Fisher Scientific, Madison, WI, USA) spectrometer in the attenuated total reflectance (ATR) mode.

X-ray diffraction (XRD)

XRD data were measured by a PANalytical X’pert PRO MRD X-ray diffractometer equipped with Cu Kα radiation (*λ* = 1.54056 Å). 

Ultraviolet−visible (UV−Vis) absorption 

Samples were characterized using a UV−Vis−NIR spectrophotometer (Lambda 750, PerKinElmer Inc., Waltham, MA, USA).

### 3.8. Mechanical Evaluation 

The elastic modulus and nanohardness of the synthesized materials were evaluated with nanoindentation testing using a Berkovich tip (G200, Agilent Technologies, Santa Clara, CA, USA). Before evaluation, the tip was calibrated with a fused silica sample. The loading and unloading times were both 15 s, and the holding time was 10 s. During evaluation, the maximum loading and unloading force applied was 0.098 N. Testworks 4 software (MTS Systems Corporation, Eden Prairie, MN, USA) was used to record and calculate the measured data. Statistical software (SPSS Statistic 24; IBM) was applied to assess and analyze the differences in the elastic modulus and nanohardness of the synthesized materials. Differences were considered significant at *p* < 0.05. The data are expressed as mean ± standard deviation.

### 3.9. Antibacterial and Cytocompatibility Evaluation

#### 3.9.1. Colony-Forming Unit (CFU) Assay 

To evaluate the antibacterial adhesion of the biofunctional materials, a CFU assay was applied to evaluate the antibacterial adhesion ability of the bioactive material. Synthesized materials (Group A with 0.2 wt% GO, *n* = 10; Group B with 0.4 wt% GO, *n* = 10; and Group C without the addition of GO, *n* = 10) with a size of 3 × 3 × 1.5 mm^3^ were prepared and autoclave-sterilized at 121 °C for 60 min (SSR-3A; Consolidated Sterilizer System, Boston, MA, USA). Samples were immersed in artificial saliva (0.33 g of KH_2_PO_4_, 0.34 g of Na_2_HPO_4_, 1.27 g of KCl, 0.16 g of NaSCN, 0.58 g of NaCl, 0.17 g of CaCl_2_, 0.16 g of NH_4_Cl, 0.2 g of urea, 0.03 g of glucose, 0.002 g of ascorbic acid, and 2.7 g of mucin in 1000 mL of distilled water, pH 7) overnight. Streptococcus mutans (ATCC 35668, ATCC, Manassas, VA, USA) cells were cultured in horse blood medium (31.2 g of Columbia agar base, 800 mL of distilled water, 40 mL of horse blood, 0.1 g of hemin, 0.08 g of NAOH, and 200 mL of distilled water) under anaerobic conditions (85% N_2_, 10% H_2_ and 5% CO_2_) at 37 °C in an anaerobic chamber (Forma Anaerobic Chamber; Thermo Fisher Scientific, Inc.). Cells were harvested by centrifugation (5000 rpm for 10 min), washed once with 10 mM sodium phosphate-buffered saline (PBS, pH 7.2), and resuspended in brain-heart infusion (BHI) broth (Difco Laboratories, Detroit, MI, USA) at a concentration of 10^6^ CFU/mL. Samples were horizontally placed at the bottom of a 96-well plate with the addition of 200 μL of S. mutans in BHI (10^6^ CFU/mL). 

After incubation under anaerobic conditions at 37 °C for 24 h, the samples were rinsed three times with PBS, placed in 1 mL of fresh BHI solution, and then treated by sonication for 60 s to collect the adherent bacteria from the sample surfaces. The procedures were conducted on a horizontal laminar flow clean bench (Polypropylene Horizontal Laminar Flow Clean Bench; AirClean 5000 Workstation, AirClean Systems, Creedmoor, NC, USA). Ten-fold serial dilutions of the bacterial suspensions were plated in duplicate on horse blood agar. After 48 h of incubation, the CFUs were counted. Three independent biofilm experiments were performed.

#### 3.9.2. Cytocompatibility Evaluation 

HGF-1 cells were obtained from the American Type Culture Collection and used to test the biosecurity of the fabricated biomaterials. HGF-1 cells were cultured in high-glucose Dulbecco’s modified Eagle’s medium (DMEM; Sigma-Aldrich) supplemented with 4 mM L-glutamine (Sigma-Aldrich) and 10% (vol/vol) heat-inactivated fetal bovine serum (FBS; Sigma-Aldrich). No antibiotic supplements were used. The cells were incubated at 37 °C in a 5% CO_2_ atmosphere, fed every 48 h, and routinely subcultured every five days with a split ratio of 1:3, using 1 × trypsin-EDTA (0.05%; Sigma-Aldrich) for 3 min at 37 °C. The synthesized materials (Group A with 0.2 wt% GO, *n* = 10; Group B with 0.4 wt% GO, *n* = 10; and Group C without the addition of GO, *n* = 10) with a size of 3 × 3 × 1.5 mm^3^ were prepared. After autoclave sterilization, the samples were placed into a 96-well plate and incubated with 1 mL of cell suspension (6.0 × 10^3^ cells/well) in a 5% CO_2_ incubator at 37 °C for 7 days. The Cell Counting Kit-8 (CCK-8; Dojindo Molecular Technology, Kumamoto, Japan) was used to quantitatively evaluate the cell viability of the samples by staining living cells. After incubation for 4 hours at 37 °C, the resultant production of water-soluble formazan dye was assayed at a wavelength of 450 nm by a microplate reader (Multiscan MK3; Thermo Fisher Scientific, Inc.).

## 4. Conclusions

We used an effective acellular method to fabricate a biofunctional material with an enamel-like structure. This biofunctional material successfully recreated the distinctive hierarchical architecture of enamel from the crystallographic to the nano-, micro-, and macro-scales. GO served as a substrate; therefore, the mechanical performance and biocompatibility of the synthesized material was similar to native enamel. The synthesized biofunctional material with 0.2 wt% GO matrix demonstrated nanohardness ((3.170 ± 0.285) GPa) and a modulus ((75.172 ± 5.014) GPa) comparable to native enamel. In addition, it demonstrated antibacterial properties and its ability to resist bacterial adhesion. After incubation for 24 h, the quantity of adherent bacteria on the synthesized material with 0.2 wt% GO matrix ((45.4 ± 21.65) × 10^4^ CFU/mL) was significantly less than that on the synthesized material without the addition of the GO matrix ((141.20 ± 47.55) × 10^4^ CFU/mL). This study offers a promising candidate for dental restoration and brings new perspectives for the synthesis of enamel-inspired biomaterials.

## Figures and Tables

**Figure 1 ijms-23-13810-f001:**
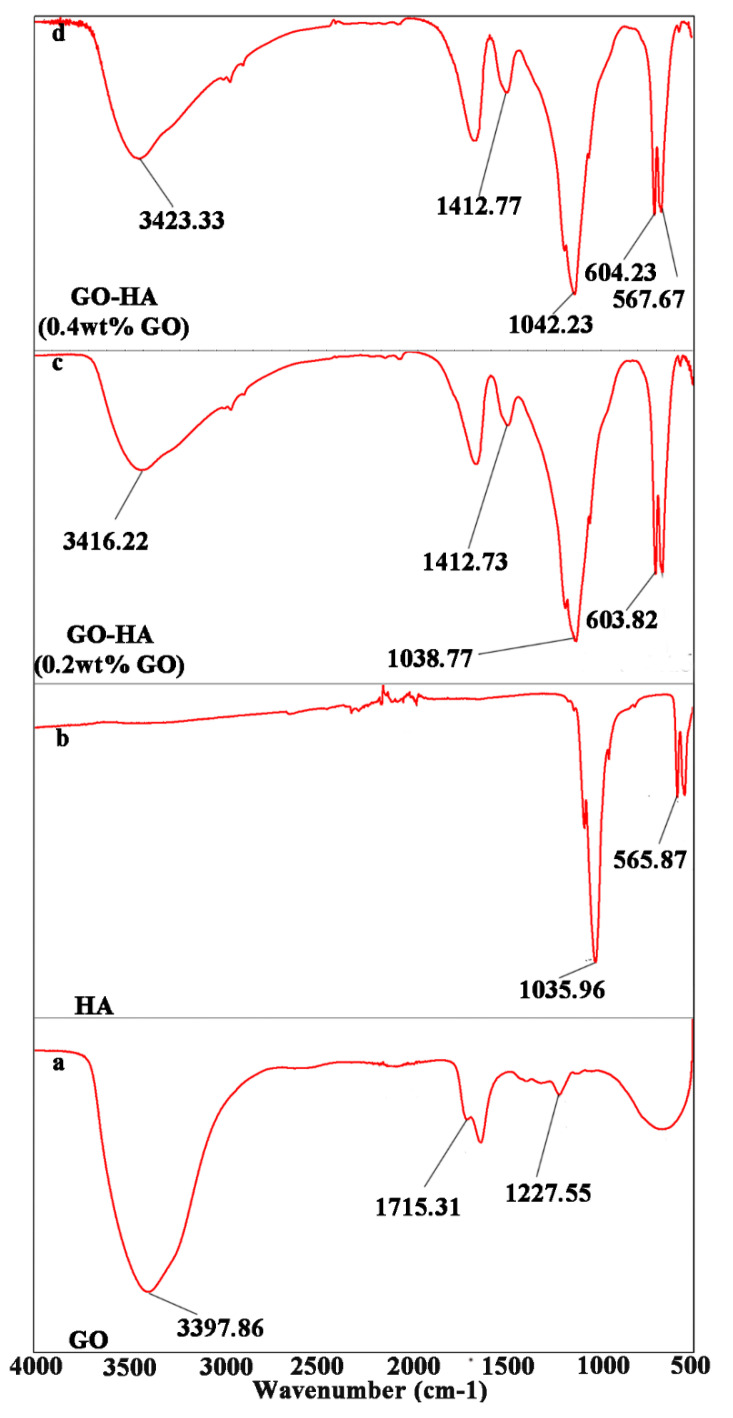
Fourier transform infrared (FTIR) spectroscopy evaluation of graphene oxide (GO; **a**), hydroxyapatite (HA; **b**), GO–HA (fabricated with 0.2 wt% GO; **c**) and GO–HA (fabricated with 0.4 wt% GO; **d**).

**Figure 2 ijms-23-13810-f002:**
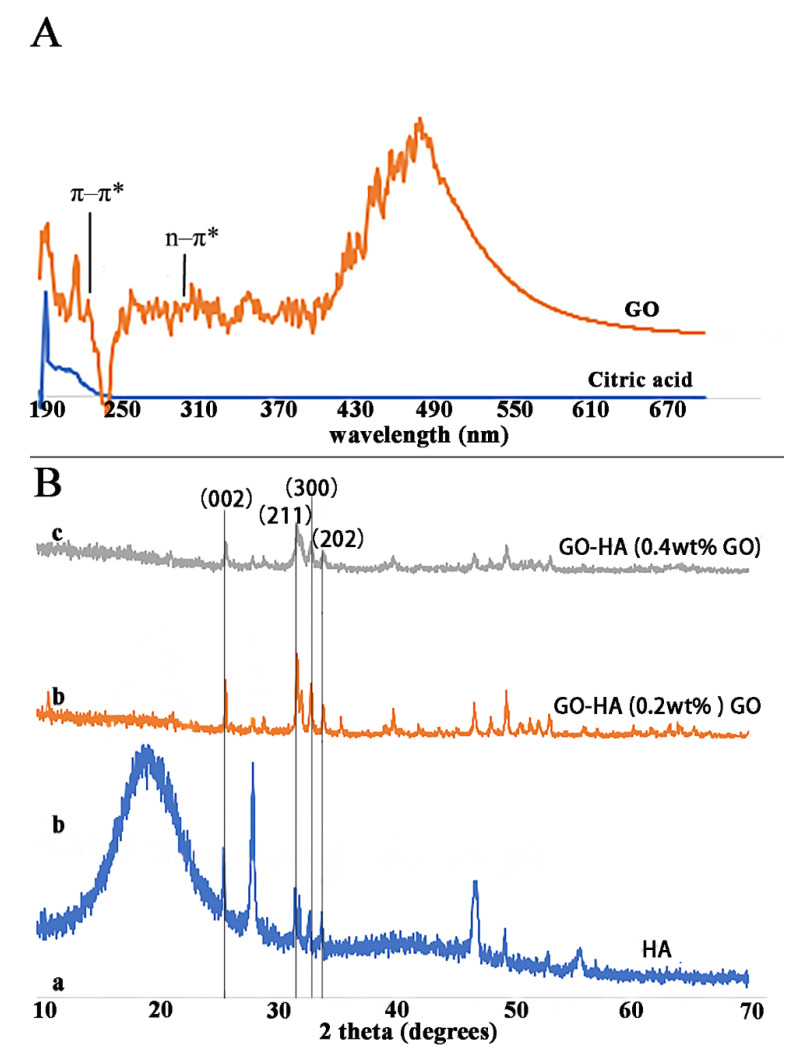
(**A**) Ultraviolet-visible (UV-visible) absorption spectra of citric acid and graphene oxide (GO); (**B**) XRD evaluation of hydroxyapatite (HA) and GO−HA crystals fabricated with different concentrations of GO.

**Figure 3 ijms-23-13810-f003:**
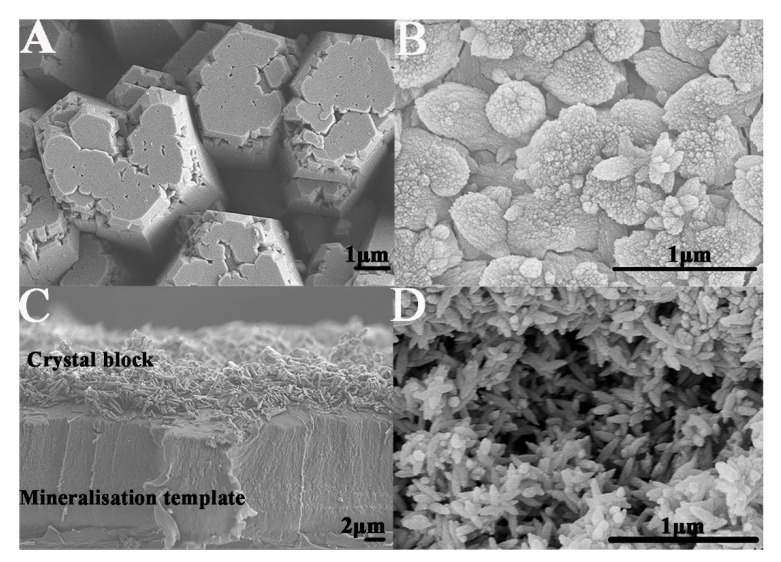
Scanning electron microscopy (SEM) evaluation. (**A**) Hydroxyapatite crystals (HA); (**B**) graphene oxide−hydroxyapatite crystals (GO−HA); (**C**,**D**) transversal and surface micrographs of GO−HA crystal block, respectively.

**Figure 4 ijms-23-13810-f004:**
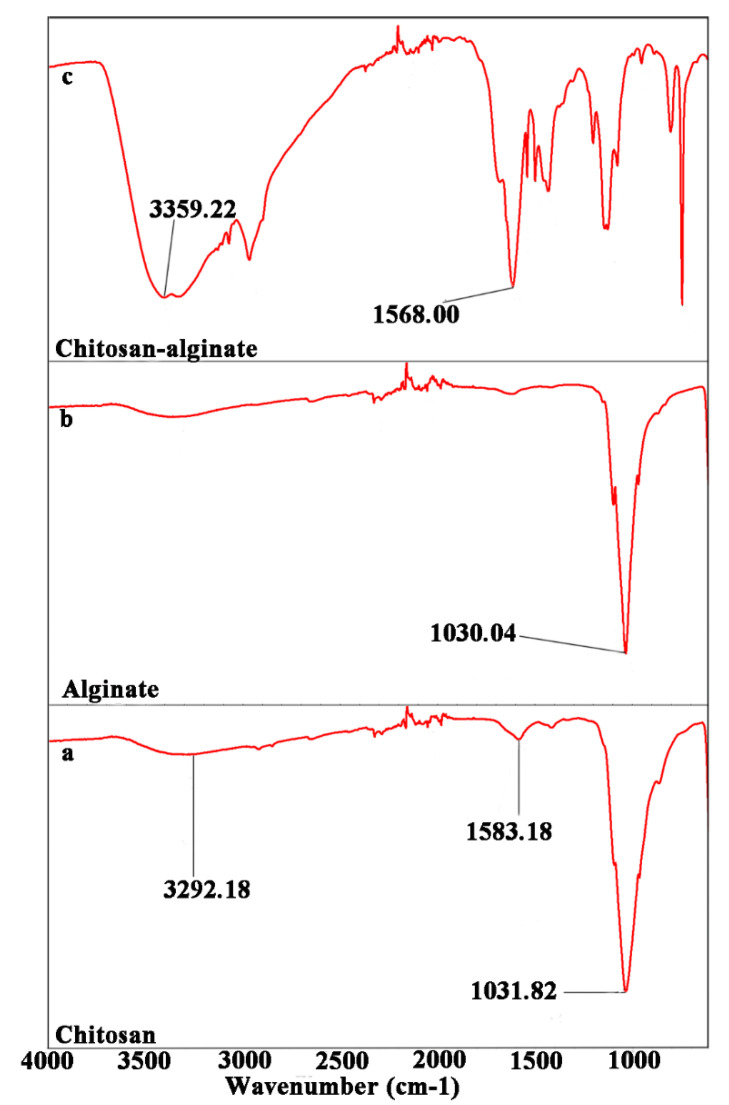
Fourier transform infrared (FTIR) spectroscopy evaluation of chitosan (**a**), alginate (**b**), and chitosan-alginate polyelectrolyte complexes (**c**).

**Figure 5 ijms-23-13810-f005:**
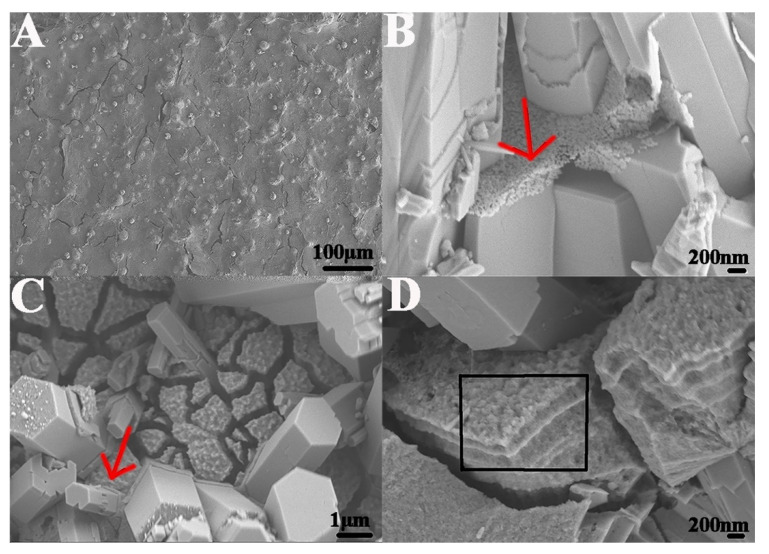
Scanning electron microscopy (SEM) evaluation of graphene oxide-hydroxyapatite (GO–HA) crystals treated with two cycles of mineralization and layer-by-layer deposition. (**A**) Image taken two days after remineralization. (**B**) Image taken three days after remineralization (the arrow represents the mineralized polyelectrolyte matrixes covered on the surface of the crystals). (**C**,**D**) Images taken four days after remineralization. The arrow in (**C**) represents the overgrowth of the original crystal, and the rectangle in (**D**) represents the mineralized polyelectrolyte matrixes in the voids among the crystals.

**Figure 6 ijms-23-13810-f006:**
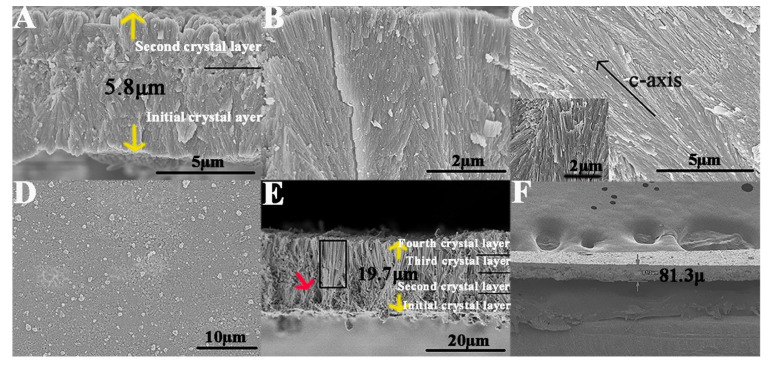
Scanning electron microscopy (SEM) evaluations of the synthesized biofunctional materials. (**A**) Transversal section of the synthesized material fabricated with two cycles of mineralization and LBL deposition. (**B**,**C**) Magnified micrographs of (**A**). The inset in (**C**) shows the prismatic structure of native enamel. (**D**) Surface micrograph of the second crystal layer in the synthesized material. (**E**) Transversal micrograph of the synthesized material fabricated with four mineralization and layer-by-layer deposition cycles. The arrow in red represents the formation of new crystals, and the rectangle represents the overgrowth of the original crystals. (**F**) Image of the final synthesized material after 10 cycles.

**Figure 7 ijms-23-13810-f007:**
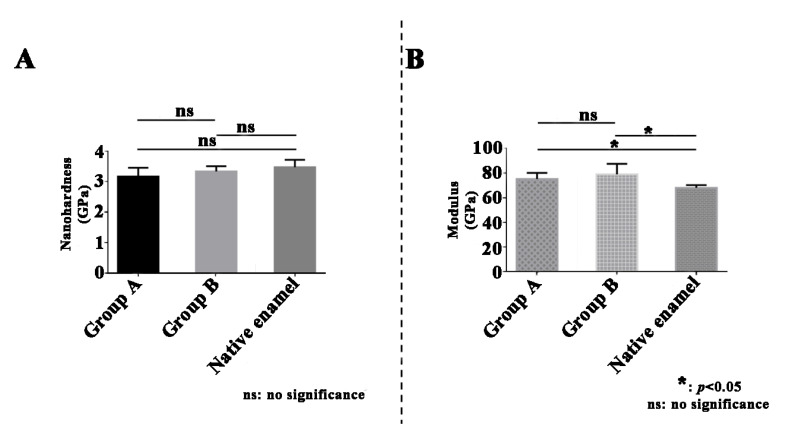
Mechanical evaluation of the synthesized materials. (**A**) Nanohardness of synthesized materials with 0.2 wt% graphene oxide (GO; Group A), with 0.4 wt% GO (Group B), and native enamel. (**B**) Elastic modulus of synthesized materials with 0.2 wt% GO (Group A), with 0.4 wt% GO (Group B), and native enamel.

**Figure 8 ijms-23-13810-f008:**
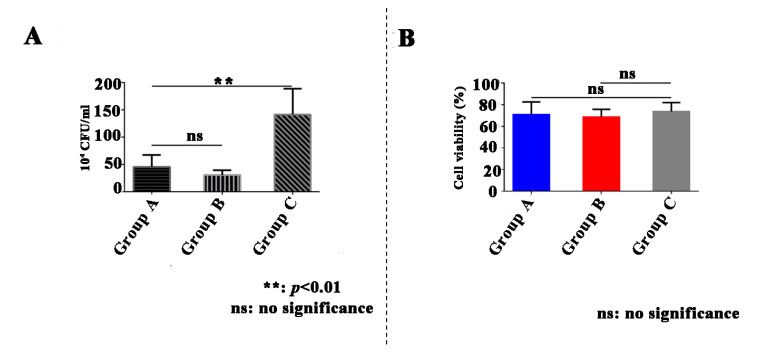
Antibacterial and cytocompatibility evaluation. (**A**) Colony-forming unit (CFU) evaluation of enamel and synthesized biofunctional materials with 0.2 wt% graphene oxide (GO; Group A), with 0.4 wt% GO (Group B), and without the addition of GO (Group C) after incubation for 24 h. (**B**) Quantity of cells proliferated on synthesized biofunctional materials with 0.2 wt% GO (Group A), with 0.4 wt% GO (Group B), and without the addition of GO (Group C) after cell culture for seven days.

**Table 1 ijms-23-13810-t001:** Concentration of carbon and oxygen in citric acid and graphene oxide (GO).

	Carbon (wt%)	Oxygen (wt%)
Citric acid	35.34	64.66
GO	49.37	50.63

**Table 2 ijms-23-13810-t002:** Concentration of different elements in pure hydroxyapatite (HA) crystals and graphene oxide-hydroxyapatite (GO−HA) crystals fabricated with different concentrations of GO.

	Carbon (wt%)	Oxygen (wt%)	Fluorine (wt%)	Phosphorous (wt%)	Calcium (wt%)
Pure HA crystal	1.83	24.51	3.19	22.08	44.39
GO−HA crystal fabricated with addition of 0.2 wt% GO	7.86	26.08	7.97	18.69	39.40
GO−HA crystal fabricated with addition of 0.4 wt% GO	9.25	26.03	7.84	18.78	38.10

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
