# Peer review of "Fabrication of Multilayered Biofunctional Material with an Enamel-like Structure"

_ijms, 2022, doi:10.3390/ijms232213810_

Round 1

Reviewer 1 Report

In References list, reference 47. says "[48] Li ...." [48] must be deleted.

Author Response

Dear Editor,

Thank you very much for considering our manuscript entitled “Fabrication of Multilayered Biofunctional Material with an Enamel-like Structure” (ijms-1993023; original title: Multilayered Biofunctional Material with an Enamel-like Structure). Please see our responses to reviewers’ comments below. Revisions in the marked copy are highlighted in red. The lines’ numbers mentioned in the responses were according to the marked copy of the manuscript.

Comments from Reviewer #1:

  1. In References list, reference 47. says "[48] Li ...." [48] must be deleted.

Response: We deleted the typo.

“47. Li, D.; Sun, H.; Jiang, L.; Zhang, K.; Liu, W.; Zhu, Y.; Fangteng, J.; Shi, C.; Zhao, L.; Sun, H.; Yang, B. Enhanced biocompatibility of PLGA nanofibers with gelatin/nano-hydroxyapatite bone biomimetics incorporation. ACS Appl Mate Inter 2014, 6, 9402-9410.” (line 615-617)

.

Reviewer 2 Report

Comments on ijms-1993023

The manuscript entitled “Multilayered Biofunctional Material with an Enamel-Like Structure” presents the method of evaporation-based bottom-up self-assembly combined with a layer-by-layer technique and introduced an antibacterial agent i.e., graphene oxide to fabricate a biofunctional material with an enamel-like architecture and antibacterial ability. Then, they fabricated the bulk macroscopic multilayered biofunctional material with hierarchical enamel-like architecture.

The manuscript has several amendments required to be resolved before accepting it for publication which are disclosed below:

·         The novelty of the presented work is missing in the Abstract. Please correct the spelling mistake of biofunctional in Abstract.

·         The resolution of the figures is very poor. Please provide all the figures in high resolution and with proper formatting.

·         The font size of all the axes of the figures is very small and can barely able to see. Please increase the font size of each figure. For example, 1, 2, 4, 7, and 8.

·         Please label all the peaks of the XRD pattern which is missing in Figure 2(b). Similarly in Figure 2(a), the data of UV–spectra are not labeled, which one is citric acid and GO??

·         Furthermore, since graphene oxide contains abundant hydroxyl and carboxyl groups as well as dangling bonds and defects, the authors didn’t discuss anything related to that, could authors discuss how the defects affect the properties of their synthesized structure?

·         Why do the authors use specifically Graphene oxide why not any other 2D materials which have almost the same properties??

·         In Figure 3, the scale bar of SEM figures is wrong, how 1 micron may have different scale bars in all the sub-figures of Figure 3. Please have a look carefully. And the authors should label the SEM images and provide images with different magnifications for better understanding. Similarly Figures 5 and 6. And it is better to use colors that are visible and easy to read.

·         Similarly, Figures 7 and 8 are not visible, it is very difficult for a reader to analyze the data.

·         The authors should provide more discussions on the mechanisms for performance strategies, which would be beneficial for readers to understand their significance.

·         The authors should provide a comparison of the mechanical properties of their synthetic enamel-like structure with other reported works.

·         The authors are suggested to write a few sentences to propose some prospective applications of their work in the last paragraph before the Conclusions.

·         The conclusions are poorly written, it is suggested to emphasize the novelty of their work and discuss the main findings/results.

·         It is also recommended to add some references from recent years of the related work. The format of the references is also not consistent.

·         The language expression in the text needs to be carefully checked and revised. There are some grammatical mistakes.

Author Response

Dear Editor,

Thank you very much for considering our manuscript entitled “Fabrication of Multilayered Biofunctional Material with an Enamel-like Structure” (ijms-1993023; original title: Multilayered Biofunctional Material with an Enamel-like Structure). Please see our responses to reviewers’ comments below. Revisions in the marked copy are highlighted in red. The lines’ numbers mentioned in the responses were according to the marked copy of the manuscript.

Comments from Reviewer #2:

The manuscript entitled “Multilayered Biofunctional Material with an Enamel-Like Structure” presents the method of evaporation-based bottom-up self-assembly combined with a layer-by-layer technique and introduced an antibacterial agent i.e., graphene oxide to fabricate a biofunctional material with an enamel-like architecture and antibacterial ability. Then, they fabricated the bulk macroscopic multilayered biofunctional material with hierarchical enamel-like architecture.

The manuscript has several amendments required to be resolved before accepting it for publication which are disclosed below:

  1. The novelty of the presented work is missing in the Abstract. Please correct the spelling mistake of biofunctional in Abstract.

Response: We have revised the spelling mistakes and rewritten the abstract.

“The oral cavity is an environment with diverse bacteria; thus, antibacterial materials are crucial for treating and preventing dental diseases. There is a high demand for materials with an enamel-like architecture because of the high failure rate of dental restorations due to the physical differences between dental materials and enamel. However, recreating the distinctive apatite composition and hierarchical architecture of enamel is challenging. The aim of this study was to synthesize a novel material with enamel-like structure and antibacterial ability. We established a non-cell biomimetic method of evaporation-based bottom-up self-assembly combined with a layer-by-layer technique and introduced an antibacterial agent (graphene oxide) to fabricate a biofunctional material with an enamel-like architecture and antibacterial ability. Specifically, enamel-like graphene oxide-hydroxyapatite crystals formed on a customized mineralization template were assembled into an enamel-like prismatic structure with a highly organized orientation preferentially along the c-axis through evaporation-based bottom-up self-assembly. With the aid of layer-by-layer absorption, we then fabricated a bulk macroscopic multilayered biofunctional material with a hierarchical enamel-like architecture. This enamel-inspired biomaterial could effectively resolve the problem in dental restoration and brings new prospects for the synthesis of other enamel-inspired biomaterials.” (line 9-23)

  1. The resolution of the figures is very poor. Please provide all the figures in high resolution and with proper formatting.

Response: All figures with the high revolution and proper format were provided.

Figure 1. Fourier-transform infrared spectroscopy evaluation of pure hydroxyapatite (HA), graphene oxide (GO), and GO–HA fabricated with different concentrations of GO.

Figure 2 (A) Ultraviolet-visible absorption spectra of citric acid and graphene oxide (GO); (B) XRD evaluation of hydroxyapatite (HA) and GO–HA crystals fabricated with different concentrations of GO.

Figure 3. Scanning electron microscopy evaluation. (A) Hydroxyapatite crystals (HA); (B) Graphene oxide-hydroxyapatite crystals (GO-HA); (C, D) Transversal and surface micrographs of GO-HA crystal block, respectively.

Figure 4. Fourier-transform infrared spectroscopy evaluation of chitosan, alginate, and chitosan–alginate polyelectrolyte complexes.

Figure 5. Scanning electron microscopy evaluation of graphene oxide-hydroxyapatite (GO-HA) crystals treated with two cycles of mineralization and layer-by-layer deposition. (A) Image taken two days after remineralization. (B) Image taken three days after remineralization (the arrow represents the mineralized polyelectrolyte matrixes covered on the surface of the crystals). (C, D) Images taken four days after remineralization. The arrow in (C) represents the overgrowth of the original crystal, and the rectangle in (D) represents the mineralized polyelectrolyte matrixes in the voids among the crystals.

Figure 6. Scanning electron microscopy evaluations of the synthesized biofunctional materials. (A) Transversal section of the synthesized material fabricated with two cycles of mineralization and LBL deposition. (B, C) Magnified micrographs of (A). The inset in (C) shows the prismatic structure of native enamel. (D) Surface micrograph of the second crystal layer in the synthesized material. (E) Transversal micrograph of the synthesized material fabricated with four mineralization and layer-by-layer deposition cycles. The arrow in red represents the formation of new crystals, and the rectangle represents the overgrowth of the original crystals. (F) Image of the final synthesized material after 10 cycles.

Figure 7. Mechanical evaluation of the synthesized materials. (A) Nanohardness of synthesized materials with 0.2 wt% graphene oxide (GO), with 0.4 wt% GO, and without the addition of GO. (B) Elastic modulus of synthesized materials with 0.2 wt% GO, with 0.4 wt% GO, and without the addition of GO.

Figure 8. Antibacterial and cytocompatibility evaluation. (A) Colony-forming unit (CFU) evaluation of enamel and synthesized biofunctional materials with 0.2 wt% graphene oxide (GO), with 0.4 wt% GO, and without the addition of GO after incubation for 24 hours. (B) Quantity of cells proliferated on synthesized biofunctional materials with 0.2 wt% GO, with 0.4 wt% GO, and without the addition of GO after cell culture for seven days.

  1. The font size of all the axes of the figures is very small and can barely able to see. Please increase the font size of each figure. For example, 1, 2, 4, 7, and 8.

Response: The axes markers in figures have been enlarged. 

Figure 1. Fourier-transform infrared spectroscopy evaluation of pure hydroxyapatite (HA), graphene oxide (GO), and GO–HA fabricated with different concentrations of GO.

Figure 2 (A) Ultraviolet-visible absorption spectra of citric acid and graphene oxide (GO); (B) XRD evaluation of hydroxyapatite (HA) and GO–HA crystals fabricated with different concentrations of GO.

Figure 4. Fourier-transform infrared spectroscopy evaluation of chitosan, alginate, and chitosan–alginate polyelectrolyte complexes.

Figure 7. Mechanical evaluation of the synthesized materials. (A) Nanohardness of synthesized materials with 0.2 wt% graphene oxide (GO), with 0.4 wt% GO, and without the addition of GO. (B) Elastic modulus of synthesized materials with 0.2 wt% GO, with 0.4 wt% GO, and without the addition of GO.

Figure 8. Antibacterial and cytocompatibility evaluation. (A) Colony-forming unit (CFU) evaluation of enamel and synthesized biofunctional materials with 0.2 wt% graphene oxide (GO), with 0.4 wt% GO, and without the addition of GO after incubation for 24 hours. (B) Quantity of cells proliferated on synthesized biofunctional materials with 0.2 wt% GO, with 0.4 wt% GO, and without the addition of GO after cell culture for seven days.

  1. Please label all the peaks of the XRD pattern which is missing in Figure 2(b). Similarly in Figure 2(a), the data of UV–spectra are not labeled, which one is citric acid and GO??

Response: In Figure 2 (a), the spectrum of citric acid and GO have been labelled, respectively. The peaks mentioned in the main text have been labelled in Figure 2 (b). 

Figure 2 (A) Ultraviolet-visible absorption spectra of citric acid and graphene oxide (GO); (B) XRD evaluation of hydroxyapatite (HA) and GO–HA crystals fabricated with different concentrations of GO.

  1. Furthermore, since graphene oxide contains abundant hydroxyl and carboxyl groups as well as dangling bonds and defects, the authors didn’t discuss anything related to that, could authors discuss how the defects affect the properties of their synthesized structure?

Response: The number of defects increased with respect to the increasing heating temperature [1]. The defects in synthesized GO may affect its mechanical properties. Since the heating temperature was relatively low (200℃) in this study, there may not be too many defects. In addition, our results of mechanical evaluation demonstrated that the mechanical properties were improved after the addition of GO.

  1. Maulana, A.; Nugraheni, A.Y.; Jayanti, D.N.; Mustofa, S.; Baqiya, M.A. Defect and magnetic properties of reduced graphene oxide prepared from old coconut shell. In IOP Conference Series: Materials Science and Engineering 2017, 196, 012021.

  1. Why do the authors use specifically Graphene oxide why not any other 2D materials which have almost the same properties??

Response: Due to its excellent mechanical properties, GO has been extensively investigated. Researchers have also started to explore the application of GO in dentistry due to its antibacterial and mechanical properties [1,2,3]. The biosafety of GO in dental application has been confirmed. Therefore, we used GO as the additive in this study.

  1. Nizami, M.Z.I.; Takashiba, S.; Nishina, Y. Graphene oxide: A new direction in dentistry. Applied Materials Today2020, 19, 100576.
  2. Qi, X.; Jiang, F.; Zhou, M.; Zhang, W.; Jiang, X. Graphene oxide as a promising material in dentistry and tissue regeneration: A review. Smart Materials in Medicine, 20212, 280-291.
  3. Yang, X.; Zhao, Q.; Chen, Y.; Fu, Y.; Lu, S., Yu, X., Yu, D. and Zhao, W., 2019. Effects of graphene oxide and graphene oxide quantum dots on the osteogenic differentiation of stem cells from human exfoliated deciduous teeth. Artificial cells, nanomedicine, and biotechnology 2019, 47, 822-832.

  1. In Figure 3, the scale bar of SEM figures is wrong, how 1 micron may have different scale bars in all the sub-figures of Figure 3. Please have a look carefully. And the authors should label the SEM images and provide images with different magnifications for better understanding. Similarly Figures 5 and 6. And it is better to use colors that are visible and easy to read.

Response: We relabeled the markers in SEM figures for better understanding. We have double checked the scale bar of SEM figures and found that they were correct. The original SEM figures are provided below for your reference. We have also used different colors accordingly.

Figure 3. Scanning electron microscopy evaluation. (A) Hydroxyapatite crystals (HA); (B) Graphene oxide-hydroxyapatite crystals (GO-HA); (C, D) Transversal and surface micrographs of GO-HA crystal block, respectively.

Figure 5. Scanning electron microscopy evaluation of graphene oxide-hydroxyapatite (GO-HA) crystals treated with two cycles of mineralization and layer-by-layer deposition. (A) Image taken two days after remineralization. (B) Image taken three days after remineralization (the arrow represents the mineralized polyelectrolyte matrixes covered on the surface of the crystals). (C, D) Images taken four days after remineralization. The arrow in (C) represents the overgrowth of the original crystal, and the rectangle in (D) represents the mineralized polyelectrolyte matrixes in the voids among the crystals.

Figure 6. Scanning electron microscopy evaluations of the synthesized biofunctional materials. (A) Transversal section of the synthesized material fabricated with two cycles of mineralization and LBL deposition. (B, C) Magnified micrographs of (A). The inset in (C) shows the prismatic structure of native enamel. (D) Surface micrograph of the second crystal layer in the synthesized material. (E) Transversal micrograph of the synthesized material fabricated with four mineralization and layer-by-layer deposition cycles. The arrow in red represents the formation of new crystals, and the rectangle represents the overgrowth of the original crystals. (F) Image of the final synthesized material after 10 cycles.

  1. Similarly, Figures 7 and 8 are not visible, it is very difficult for a reader to analyze the data.

Response: We have relabeled the markers in Figure 7 and 8 for better understanding.

Figure 7. Mechanical evaluation of the synthesized materials. (A) Nanohardness of synthesized materials with 0.2 wt% graphene oxide (GO), with 0.4 wt% GO, and without the addition of GO. (B) Elastic modulus of synthesized materials with 0.2 wt% GO, with 0.4 wt% GO, and without the addition of GO.

Figure 8. Antibacterial and cytocompatibility evaluation. (A) Colony-forming unit (CFU) evaluation of enamel and synthesized biofunctional materials with 0.2 wt% graphene oxide (GO), with 0.4 wt% GO, and without the addition of GO after incubation for 24 hours. (B) Quantity of cells proliferated on synthesized biofunctional materials with 0.2 wt% GO, with 0.4 wt% GO, and without the addition of GO after cell culture for seven days.

  1. The authors should provide more discussions on the mechanisms for performance strategies, which would be beneficial for readers to understand their significance.

Response: We have added more discussion on the mechanism for performance strategies.

In this study, we established a non-cell-based biomimetic strategy to fabricate a novel bioactive material to resolve the limitations of the conventional dental materials. The material owns enamel-like structural, mechanical, and esthetic properties, as well as antibacterial abilities and biocompatibility.

The oral cavity is an environment that pathogenesis and development of diseases are known to happen frequently. The physical and structural mismatch between dental restorative materials and native enamel can cause marginal leakage, resulting in recurrent caries and restoration failure. Dental enamel is characterized by its excellent intrinsic-extrinsic fracture toughening mechanism, due to its organized hierarchical structure. Synthesis of the biomaterial with enamel-like structure and anti-bacterial ability may effectively resolve the problems in dental restoration. In amelogenesis, enamel organic matrixes play the role of template to guide the minerals into parallelly aligned HA prisms with highly organized orientation [32,38]. Inspired by this process of native enamel formation, the templated-directed method has been proposed and applied to synthesize materials with a highly ordered orientation [39-41]. Designing an epitaxial template is arduous in the templated-directed method due to the stringent requirements for the template surface structure. Epitaxial templates should have rigid and highly ordered surface structures for imposing lattice matching [42]. In addition, removing the template to obtain individual nanomaterials is challenging.

Our solution to this challenge was to establish a non-cell-based biomimetic strategy by using a customized polyethylene membrane coated with polydopamine. Polyethylene membranes are soluble in isoamyl acetate solution. Polydopamine has strong adhesive properties to various substrates, and its catecholamine moieties can bind with Ca2+, further attracting PO43− and triggering crystal nucleation [43,44]. HA formed with the aid of polydopamine is aligned to the c-axis and parallel to the polydopamine layer [45]. Those enamel-like GO-HA crystals were continuing to aggregate and assemble into a crystal layer with the highly organized crystal orientation. The macroscopic, multilayered GO–HA crystal was then constructed via sequential growth of the GO–HA crystal layers, followed by LBL deposition of a polyelectrolyte matrix. Lastly, we isolated the bulk GO–HA crystal and obtained the synthesized material by dissolving the mineralization template in an isoamyl acetate solution.” (line 86-118)

  1. The authors should provide a comparison of the mechanical properties of their synthetic enamel-like structure with other reported works.

Response: We have added the discussion of comparison of mechanical properties with another enamel-inspired material.

“Wei et al. also synthesized a multilayered composite with enamel-like structure which has hardness and ultrahigh Young’s modulus comparable to natural enamel [61]. However, this multilayered composite is composed of rutile dioxide nanorods that are different from the main constituent of native enamel.” (line 326-330)

“61. Wei, J., Ping, H., Xie, J., Zou, Z., Wang, K., Xie, H., Wang, W., Lei, L., Fu, Z. Bioprocess‐inspired microscale additive manufacturing of multilayered TiO2/polymer composites with enamel‐like structures and high mechanical properties. Adv Funct Mater 2020, 30; 1904880.” (line 646-648 )

  1. The authors are suggested to write a few sentences to propose some prospective applications of their work in the last paragraph before the Conclusions.

Response: We have proposed some prospective applications of our work in the last paragraph before the Conclusion.

We used an effective acellular method to fabricate a macroscopic multilayered GO-HA material with a hierarchical enamel-like architecture and antibacterial abilities. This enamel-inspired biofunctional material recreated the distinctive hierarchical architecture of enamel from crystallographic scale to the nano-, micro- and macroscale. Its compositions are similar to that of native enamel. This study offers a promising candidate for dental restoration. The computer-aided design/computer-aided manufacturing (CAD/CAM) technique can also be applied to form the exact morphology and size of the restoration needed before it is inserted into the prepared cavity of the tooth.” (line 355-362)

  1. The conclusions are poorly written, it is suggested to emphasize the novelty of their work and discuss the main findings/results.

Response: We have rewritten the conclusion.

We used an effective acellular method to fabricate a biofunctional material with an enamel-like structure. This biofunctional material successfully recreated the distinctive hierarchical architecture of enamel from the crystallographic to the nano-, micro-, and macro-scales. It has the mechanical performance and biocompatibility that are comparable to native enamel. In addition, it has the antibacterial ability to resist bacterial adhesion. This study offers a promising candidate for dental restoration and brings new perspectives for the synthesis of enamel-inspired biomaterials.” (line 496-502)

  1. It is also recommended to add some references from recent years of the related work. The format of the references is also not consistent.

Response: We have added some recent references and checked the format of all references.

“5. Dorozhkin, S.V. 2010. Nanosized and nanocrystalline calcium orthophosphates. Acta Biomater 20106, 715-734.” (line 523)

“8. Bakry, A.S.; Takahashi, H.; Otsuki, M.; Tagami, J. Evaluation of new treatment for incipient enamel demineralization using 45S5 bioglass. Dent Mater 2014, 30, 314-320.” (line 528-529)

“10. Paradowska-Stolarz, A.; Wieckiewicz, M.; Owczarek, A, Wezgowiec, J. Natural Polymers for the Maintenance of Oral Health: Review of Recent Advances and Perspectives. Int J Mol Sci 2021, 25; 22; 10337.” (Line 532-533)

“11. Cox, D.; Brennan, M.; Moran, N. Integrins as therapeutic targets: lessons and opportunities. Nat Rev Drug Discov 20109, 804-820.” (line 534-535)

“13. Istikharoh, F.; Sujuti, H.; Mustamsir, E.; Swastirani, A. Preparation and biodegradable properties of hydroxyapatite nanoparticle composite coated with poly lactic-co-glycolic acid/polyvinyl alcohol for bone regeneration. Dent Med Probl 2020;57, 363–367.” (line 538-540)

“16. Ceballos-Jiménez, A.Y.; Rodríguez-Vilchis, L.E.; Contreras-Bulnes, R.; Alatorre, J.Á.A.; Velazquez-Enriquez, U.; García-Fabila, M.M. Acid resistance of dental enamel treated with remineralizing agents, Er: YAG laser and combined treatments. Dental and Medical Problems 2018, 55; 255–259.” (line 545-547)

“59. Fu, Y.C., Chen, C.H.; Wang, C.Z.; Wang, Y.H.; Chang, J.K.; Wang, G.J.; Ho, M.L.; Wang, C.K. Preparation of porous bioceramics using reverse thermo-responsive hydrogels in combination with rhBMP-2 carriers: In vitro and in vivo evaluation.J Mech Behav Biomed Mat 2013, 27, 64-76.”(line 641-643)

“61. Wei, J.; Ping, H.; Xie, J.; Zou, Z.; Wang, K.; Xie, H.; Wang, W.; Lei, L.; Fu, Z. Bioprocess‐inspired microscale additive manufacturing of multilayered TiO2/polymer composites with enamel‐like structures and high mechanical properties. Adv Funct Mater 2020, 30; 1904880.” (line 647-648)

  1. The language expression in the text needs to be carefully checked and revised. There are some grammatical mistakes.

Response: An native speaker has helped us to edit the manuscript.

Reviewer 3 Report

I read this manuscript with a real pleasure because this method can be an alternative to manufacture the artificial enamel for clinical applications in future. However I found a few flaws which have to be corrected before acceptance:

1. Abstract is very weak. Therefore, Authors have to write a new abstract contains the following sections: Background, Aim of the study, Methods, Results, Conclusions.

2. Authors cited a lot of old literature. Authors have to remove all references published before 2010. I recommend to use modern literature related to the topic.

3. Introduction: Authors have to present the latest and outstanding articles related to the topic and then write why their study is new and important. I recommend to define a clear aim of the study at the end of Introduction. I suggest to add the following outstanding literature related to the topic:

Istikharoh F, Sujuti H, Mustamsir E, Swastirani A. Preparation and biodegradable properties of hydroxyapatite nanoparticle composite coated with poly lactic-co-glycolic acid/polyvinyl alcohol for bone regeneration. Dent Med Probl. 2020;57(4):363–367. doi:10.17219/dmp/125775

Paradowska-Stolarz A, Wieckiewicz M, Owczarek A, Wezgowiec J. Natural Polymers for the Maintenance of Oral Health: Review of Recent Advances and Perspectives. Int J Mol Sci. 2021 Sep 25;22(19):10337. doi: 10.3390/ijms221910337.

Alma Y. Ceballos-Jiménez, Laura E. Rodríguez-Vilchis, Rosalía Contreras-Bulnes, Jesús Á. Arenas Alatorre, Ulises Velazquez-Enriquez, Maria M. García-Fabila. Acid resistance of dental enamel treated with remineralizing agents, Er:YAG laser and combined treatments.  Dental and Medical Problems, 2018, vol. 55, nr 3, July-September, 255–259, doi: 10.17219/dmp/93960

4. I think that Authors should add a flowchart which presents step by step the process how Authors did replicate the hierarchical architecture of enamel (it can be a graphical abstract for this manuscript).

5. Authors have to remember that title, aim of study and conclusions have to correspond one to each other. Therefore I suggest to use the following title: Novel method of multilayered biofunctional material with an enamel-like structure manufacturing

Authors have to define the clear aim and then formulate a conclusions which will be the response for the defined aim both abstract and manuscript body.

6. Please remember to use full term before first use of each abbreviation.

7. Authors have to add a legend of used abbreviations below each table and figure.

Author Response

Dear Editor,

Thank you very much for considering our manuscript entitled “Fabrication of Multilayered Biofunctional Material with an Enamel-like Structure” (ijms-1993023; original title: Multilayered Biofunctional Material with an Enamel-like Structure). Please see our responses to reviewers’ comments below. Revisions in the marked copy are highlighted in red. The lines’ numbers mentioned in the responses were according to the marked copy of the manuscript.

Comments from the Reviewer #3:

 read this manuscript with a real pleasure because this method can be an alternative to manufacture the artificial enamel for clinical applications in future. However I found a few flaws which have to be corrected before acceptance:

  1. Abstract is very weak. Therefore, Authors have to write a new abstract contains the following sections: Background, Aim of the study, Methods, Results, Conclusions.

Response: We have rewritten the abstract.

“The oral cavity is an environment with diverse bacteria; thus, antibacterial materials are crucial for treating and preventing dental diseases. There is a high demand for materials with an enamel-like architecture because of the high failure rate of dental restorations due to the physical differences between dental materials and enamel. However, recreating the distinctive apatite composition and hierarchical architecture of enamel is challenging. The aim of this study was to synthesize a novel material with enamel-like structure and antibacterial ability. We established a non-cell biomimetic method of evaporation-based bottom-up self-assembly combined with a layer-by-layer technique and introduced an antibacterial agent (graphene oxide) to fabricate a biofunctional material with an enamel-like architecture and antibacterial ability. Specifically, enamel-like graphene oxide-hydroxyapatite crystals formed on a customized mineralization template were assembled into an enamel-like prismatic structure with a highly organized orientation preferentially along the c-axis through evaporation-based bottom-up self-assembly. With the aid of layer-by-layer absorption, we then fabricated a bulk macroscopic multilayered biofunctional material with a hierarchical enamel-like architecture. This enamel-inspired biomaterial could effectively resolve the problem in dental restoration and brings new prospects for the synthesis of other enamel-inspired biomaterials.” (Line 9-23)

  1. Authors cited a lot of old literature. Authors have to remove all references published before 2010. I recommend to use modern literature related to the topic.

Response: All the old references have been replaced by recent related ones.

  1. Introduction: Authors have to present the latest and outstanding articles related to the topic and then write why their study is new and important. I recommend to define a clear aim of the study at the end of Introduction. I suggest to add the following outstanding literature related to the topic:

Istikharoh F, Sujuti H, Mustamsir E, Swastirani A. Preparation and biodegradable properties of hydroxyapatite nanoparticle composite coated with poly lactic-co-glycolic acid/polyvinyl alcohol for bone regeneration. Dent Med Probl. 2020;57(4):363–367. doi:10.17219/dmp/125775

Paradowska-Stolarz A, Wieckiewicz M, Owczarek A, Wezgowiec J. Natural Polymers for the Maintenance of Oral Health: Review of Recent Advances and Perspectives. Int J Mol Sci. 2021 Sep 25;22(19):10337. doi: 10.3390/ijms221910337.

Alma Y. Ceballos-Jiménez, Laura E. Rodríguez-Vilchis, Rosalía Contreras-Bulnes, Jesús Á. Arenas Alatorre, Ulises Velazquez-Enriquez, Maria M. García-Fabila. Acid resistance of dental enamel treated with remineralizing agents, Er:YAG laser and combined treatments.  Dental and Medical Problems, 2018, vol. 55, nr 3, July-September, 255–259, doi: 10.17219/dmp/93960

Response: The novelty and aim of this study have been added in the introduction. Suggested references have been cited in the related content.

“To overcome the shortcomings of these previous attempts, we established a method based on a combination of layer-by-layer (LBL) deposition and evaporation-based bottom-up self-assembly. LBL deposition is a simple and efficient technique commonly applied to synthesize solid films with a controlled multilayer structure, whereby assembly can be based on electrostatic interactions, hydrogen bonding, charge transfer, covalent bonding, biological recognition, or hydrophobic interactions [33,34]. The bottom-up approach is an innovative and effective approach for fabricating bulk materials. In this technique, the physical or chemical interactions between particles are used to manipulate small building blocks to construct mesoscopic or macroscopic functional architectures with well-defined morphologies, shapes, and patterns [35,36]. Evaporation has been shown to further improve the bottom-up self-assembly of enamel-like crystals [37]. By combining evaporation-based bottom-up and LBL assembly in the present study, we were able to replicate the hierarchical architecture of enamel. The aim of this study was therefore to synthesize a novel material with enamel-like structure and antibacterial ability to resolve the limitations of present dental materials.” (line 69-83)

“5. Dorozhkin, S.V. 2010. Nanosized and nanocrystalline calcium orthophosphates. Acta Biomater 20106, 715-734.” (line 523)

“8. Bakry, A.S.; Takahashi, H.; Otsuki, M.; Tagami, J. Evaluation of new treatment for incipient enamel demineralization using 45S5 bioglass. Dent Mater 2014, 30, 314-320.” (line 528-529)

“10. Paradowska-Stolarz, A.; Wieckiewicz, M.; Owczarek, A, Wezgowiec, J. Natural Polymers for the Maintenance of Oral Health: Review of Recent Advances and Perspectives. Int J Mol Sci 2021, 25; 22; 10337.” (Line 532-533)

“11. Cox, D.; Brennan, M.; Moran, N. Integrins as therapeutic targets: lessons and opportunities. Nat Rev Drug Discov 20109, 804-820.” (line 534-535)

“13. Istikharoh, F.; Sujuti, H.; Mustamsir, E.; Swastirani, A. Preparation and biodegradable properties of hydroxyapatite nanoparticle composite coated with poly lactic-co-glycolic acid/polyvinyl alcohol for bone regeneration. Dent Med Probl 2020;57, 363–367.” (line 538-540)

“16. Ceballos-Jiménez, A.Y.; Rodríguez-Vilchis, L.E.; Contreras-Bulnes, R.; Alatorre, J.Á.A.; Velazquez-Enriquez, U.; García-Fabila, M.M. Acid resistance of dental enamel treated with remineralizing agents, Er: YAG laser and combined treatments. Dental and Medical Problems 2018, 55; 255–259.” (line 545-547)

“59. Fu, Y.C., Chen, C.H.; Wang, C.Z.; Wang, Y.H.; Chang, J.K.; Wang, G.J.; Ho, M.L.; Wang, C.K. Preparation of porous bioceramics using reverse thermo-responsive hydrogels in combination with rhBMP-2 carriers: In vitro and in vivo evaluation.J Mech Behav Biomed Mat 2013, 27, 64-76.”(line 641-643)

“61. Wei, J.; Ping, H.; Xie, J.; Zou, Z.; Wang, K.; Xie, H.; Wang, W.; Lei, L.; Fu, Z. Bioprocess‐inspired microscale additive manufacturing of multilayered TiO2/polymer composites with enamel‐like structures and high mechanical properties. Adv Funct Mater 2020, 30; 1904880.” (line 647-648)

  1. I think that Authors should add a flowchart which presents step by step the process how Authors did replicate the hierarchical architecture of enamel (it can be a graphical abstract for this manuscript).

Response: A graphic abstract was included in the original submission. Maybe it is not shown in the reviewer’s version. The graphical abstract shows the process of fabrication of recreation of enamel-like structure.

“Graphic abstract: The schematic diagram of the fabrication process of the biofunctional material.”

  1. Authors have to remember that title, aim of study and conclusions have to correspond one to each other. Therefore I suggest to use the following title: Novel method of multilayered biofunctional material with an enamel-like structure manufacturing

Response: The title has been changed to “Fabrication of multilayered biofunctional Material with an Enamel-like structure”

  1. Authors have to define the clear aim and then formulate a conclusions which will be the response for the defined aim both abstract and manuscript body.

Response: The conclusion has been rewritten.

We used an effective acellular method to fabricate a biofunctional material with an enamel-like structure. This biofunctional material successfully recreated the distinctive hierarchical architecture of enamel from the crystallographic to the nano-, micro-, and macro-scales. It has the mechanical performance and biocompatibility that are comparable to native enamel. In addition, it has the antibacterial ability to resist bacterial adhesion. This study offers a promising candidate for dental restoration and brings new perspectives for the synthesis of enamel-inspired biomaterials.” (Line 496-502)

  1. Please remember to use full term before first use of each abbreviation.

Response: We have checked this accordingly.

  1. Authors have to add a legend of used abbreviations below each table and figure.

 Response: The full terms of abbreviations have been added in the captions of figures and tables.

Table 1. Concentration of carbon and oxygen in citric acid and graphene oxide (GO).” (line 140)

“Figure 1. Fourier-transform infrared (FTIR) spectroscopy evaluation of pure hydroxyapatite (HA), graphene oxide (GO), and GO–HA fabricated with different concentrations of GO.” (line 139-141)

“Figure 2 (A) Ultraviolet-visible (UV-visible) absorption spectra of citric acid and graphene oxide (GO); (B) XRD evaluation of hydroxyapatite (HA) and GO–HA crystals fabricated with different concentrations of GO.” (line 167-169)

Table 2. Concentration of different elements in pure hydroxyapatite (HA) crystals and graphene oxide-hydroxyapatite (GO–HA) crystals fabricated with different concentrations of GO.” (line 187-189)

“Figure 3. Scanning electron microscopy (SEM) evaluation. (A) Hydroxyapatite crystals (HA); (B) Graphene oxide-hydroxyapatite crystals (GO-HA); (C, D) Transversal and surface micrographs of GO-HA crystal block, respectively.” (line 210-212)

“Figure 4. Fourier-transform infrared (FTIR) spectroscopy evaluation of chitosan, alginate, and chitosan–alginate polyelectrolyte complexes.” (line 239-240)

“Figure 5. Scanning electron microscopy (SEM) evaluation of graphene oxide-hydroxyapatite (GO-HA) crystals treated with two cycles of mineralization and layer-by-layer deposition. (A) Image taken two days after remineralization. (B) Image taken three days after remineralization (the arrow represents the mineralized polyelectrolyte matrixes covered on the surface of the crystals). (C, D) Images taken four days after remineralization. The arrow in (C) represents the overgrowth of the original crystal, and the rectangle in (D) represents the mineralized polyelectrolyte matrixes in the voids among the crystals.” (line 255-261)

“Figure 6. Scanning electron microscopy (SEM) evaluations of the synthesized biofunctional materials. (A) Transversal section of the synthesized material fabricated with two cycles of mineralization and LBL deposition. (B, C) Magnified micrographs of (A). The inset in (C) shows the prismatic structure of native enamel. (D) Surface micrograph of the second crystal layer in the synthesized material. (E) Transversal micrograph of the synthesized material fabricated with four mineralization and layer-by-layer deposition cycles. The arrow in red represents the formation of new crystals, and the rectangle represents the overgrowth of the original crystals. (F) Image of the final synthesized material after 10 cycles.” (line 264-272)

Figure 7. Mechanical evaluation of the synthesized materials. (A) Nanohardness of synthesized materials with 0.2 wt% graphene oxide (GO), with 0.4 wt% GO, and without the addition of GO. (B) Elastic modulus of synthesized materials with 0.2 wt% GO, with 0.4 wt% GO, and without the addition of GO.” (line 306-309)

“Figure 8. Antibacterial and cytocompatibility evaluation. (A) Colony-forming unit (CFU) evaluation of enamel and synthesized biofunctional materials with 0.2 wt% graphene oxide (GO), with 0.4 wt% GO, and without the addition of GO after incubation for 24 hours. (B) Quantity of cells proliferated on synthesized biofunctional materials with 0.2 wt% GO, with 0.4 wt% GO, and without the addition of GO after cell culture for seven days.” (line 312-316)

Round 2

Reviewer 2 Report

Since the labeling of the x-axis is really long, the x-axis of Figures 7 and 8 can be separately defined and use some numeric values of alphabets to show in the x-axis.

Please use the stacking function for Figure 2B, the graphs shouldn't overlap. It is not convenient for a reader to understand.

Similarly, it is just a suggestion the authors can draw Figure 2 vertically way since the figure is going out of boundaries. Not looking nice

Please check Figure 5, the figure seems small compared to Figure 3. Please keep the format consistent.

The authors need to label a,b,c, and d in the figure caption as wellsince you are showing GO as A, but writing pure HO first in the figure caption (Figure 1).

Please keep the figures large enough for better understanding. Some SEM images are very small in dimension.

Conclusions should contain numerical data/values.

Author Response

Comments from Reviewer #2:

  1. Since the labeling of the x-axis is really long, the x-axis of Figures 7 and 8 can be separately defined and use some numeric values of alphabets to show in the x-axis.

Response: The synthesized materials with 0.2 wt% GO, 0.4 wt% GO and without the addition of GO were named as Group A, Group B and Group C, respectively. The x-axis of Figure 7 and 8 were changed, according to the new labels.

“Synthesized materials (Group A with 0.2 wt% GO, n = 10; Group B with 0.4 wt% GO, n = 10; and Group C without the addition of GO, n = 10) with a size of 3x3x1.5 mm3 were prepared and autoclave sterilized at 121 °C for 60 minutes (SSR-3A; Consolidated Sterilizer System, CSS).” (line 457-460)

“The synthesized materials (Group A with 0.2 wt% GO, n = 10; Group B with 0.4 wt% GO, n = 10; and Group C without the addition of GO, n = 10) with a size of 3x3x1.5 mm3 were prepared.” (line 491-493)

Figure 7. Mechanical evaluation of the synthesized materials. (A) Nanohardness of synthesized materials with 0.2 wt% graphene oxide (GO; Group A), with 0.4 wt% GO (Group B), and native enamel. (B) Elastic modulus of synthesized materials with 0.2 wt% GO (Group A), with 0.4 wt% GO (Group B), native enamel. (line 307-310)

Figure 8. Antibacterial and cytocompatibility evaluation. (A) Colony-forming unit (CFU) evaluation of enamel and synthesized biofunctional materials with 0.2 wt% graphene oxide (GO; Group A), with 0.4 wt% GO (Group B), and without the addition of GO (Group C) after incubation for 24 hours. (B) Quantity of cells proliferated on synthesized biofunctional materials with 0.2 wt% GO (Group A), with 0.4 wt% GO (Group B), and without the addition of GO (Group C) after cell culture for seven days. (line 315-320)

  1. Please use the stacking function for Figure 2B, the graphs shouldn't overlap. It is not convenient for a reader to understand. Similarly, it is just a suggestion the authors can draw Figure 2 vertically way since the figure is going out of boundaries. Not looking nice

Response: We modified the Figure 2, according to reviewer’s suggestion.

  1. Please check Figure 5, the figure seems small compared to Figure 3. Please keep the format consistent. Please keep the figures large enough for better understanding. Some SEM images are very small in dimension.

Response: We enlarged Figure 5 and kept all the figures in a consistent format.

  1. The authors need to label a,b,c, and d in the figure caption as wellsince you are showing GO as A, but writing pure HO first in the figure caption (Figure 1).

Response: We rewrote the caption of Figure 1.

Figure 1. Fourier-transform infrared (FTIR) spectroscopy evaluation of graphene oxide (GO; a), hydroxyapatite (HA; b), GO-HA (fabricated with 0.2 wt%GO; c) and GO-HA (fabricated with 0.4 wt% GO; d).” (line 139-141)

  1. Conclusions should contain numerical data/values.

Response: We added the numerical data in conclusion.

“We used an effective acellular method to fabricate a biofunctional material with an enamel-like structure. This biofunctional material successfully recreated the distinctive hierarchical architecture of enamel from the crystallographic to the nano-, micro-, and macro-scales. GO served as a substrate, leading to the mechanical performance and biocompatibility of synthesized material similar to native enamel. The synthesized biofunctional material with 0.2 wt% GO matrix had nanohardness ((3.170 ± 0.285) GPa) and modulus ((75.172 ± 5.014) GPa) comparable to native enamel. In addition, it had the antibacterial ability to resist bacterial adhesion. After incubation for 24 hours, the quantity of adherent bacteria on the synthesized material with 0.2 wt% GO matrix ((45.4 ± 21.65)x104 CFU/ml) was significantly less than that on the synthesized material without the addition of GO matrix ((141.20 ± 47.55)x104 CFU/ml). This study offers a promising candidate for dental restoration and brings new perspectives for the synthesis of enamel-inspired biomaterials.” (line 502-514)
